# Case Report: A Multi-Peptide Vaccine Targeting Individual Somatic Mutations Induces Tumor Infiltration of Neoantigen-Specific T Cells in a Patient with Metastatic Colorectal Cancer

**DOI:** 10.3390/vaccines13090960

**Published:** 2025-09-11

**Authors:** Armin Rabsteyn, Henning Zelba, Borong Shao, Lisa Oenning, Christina Kyzirakos, Simone Kayser, Tabea Riedlinger, Johannes Harter, Magdalena Feldhahn, Dirk Hadaschik, Florian Battke, Veit Scheble, Alfred Königsrainer, Saskia Biskup

**Affiliations:** 1Zentrum für Humangenetik Tübingen, 72076 Tübingen, Germanysaskia.biskup@cegat.de (S.B.); 2CeGaT GmbH, 72076 Tübingen, Germany; 3Cecava GmbH, 72076 Tübingen, Germany; 4MVZ für Diagnostik, Prävention, Onkologie und Gastroenterologie Tübingen GmbH, 72076 Tübingen, Germany; 5Allgemeine, Viszeral- und Transplantationschirurgie, Universitätsklinikum Tübingen, 72076 Tübingen, Germany

**Keywords:** personalized peptide vaccination, metastatic colorectal cancer, tumor-infiltrating lymphocytes, neoantigens, TCRβ sequencing

## Abstract

Background/Objectives: Fully personalized peptide vaccines targeting tumor-specific mutations are a promising treatment option for patients in an adjuvant but also advanced/metastatic disease situation in addition to non-personalized standard therapies. Here, we report a patient’s case with advanced metastatic colorectal cancer (mCRC) who was treated with a neoantigen-derived multi-peptide vaccine in addition to standard of care. Methods: Tumor-specific mutations were identified by whole exome and transcriptome sequencing. An individualized peptide vaccine was designed using an in-house developed epitope prediction and vaccine design platform. In this case, the vaccine consisted of 20 peptides targeting 18 distinct mutations. The vaccine was administered according to a prime-boost scheme for a total of 12 vaccinations. Vaccine immunogenicity was determined by stimulation of patient T cells with vaccinated peptides and subsequent intracellular cytokine staining (ICS). Tumor-infiltrating lymphocytes (TIL) were analyzed by ICS and T cell receptor beta chain (TCRβ) sequencing. Results: The patient survived for 41 months since initial diagnosis despite continuous disease progression under all therapeutic interventions. The vaccination induced multiple neoantigen-specific T cell responses in the patient without notable side effects. Two liver metastases were resected five months after the start of vaccination, and TIL were extracted and cultured. Analysis of TIL cultures revealed tumor infiltration by vaccine-induced neoantigen-specific T cells in only one of the metastases. TCRβ sequencing of neoantigen-specific T cells and tumor tissues supported this finding. Vaccine-targeted variants were reduced or absent in the metastasis with vaccine-specific T cell infiltration. Conclusions: This case demonstrates immunogenicity of a neoantigen-derived peptide vaccine and highlights tumor-infiltrating capabilities and potential cytotoxicity of vaccine-induced T cells in mCRC.

## 1. Introduction

Colorectal cancer (CRC) is the third most commonly diagnosed cancer worldwide and the second most common cause of cancer-related mortality [1]. CRC is projected to become increasingly relevant in the future as global incidence rates are constantly rising [2]. Despite improved screening and treatment modalities, about 25% of all primary CRC diagnoses present already as metastasized disease (mCRC), which accounts for the majority of CRC-related deaths. Due to the significantly reduced survival probability with median OS rates of 19 months and 5-year OS rates of 15%, there is still a high medical need for novel treatment approaches for mCRC [1,2]. Current treatment recommendations for mCRC vary depending on initial diagnosis and molecular subtype. Double or triple chemotherapy (FOLFOX/FOLFIRI or FOLFIRINOX) combined with anti-vascular endothelial growth factor (VEGF) antibody Bevacizumab is indicated for primary unresectable, microsatellite-stable (MSS), or proficient mismatch repair (pMMR) *RAS* wild-type mCRC. Checkpoint inhibition (CPI) using Pembrolizumab is indicated in microsatellite-instable (MSIhigh) or mismatch repair deficient (MMRd) mCRC. Anti-epidermal growth factor receptor (EGFR) therapy is indicated in left-sided *RAS* wild-type mCRC. Encorafenib plus Cetuximab is indicated in *BRAF* V600E-mutant mCRC. Surgery with/without perioperative chemotherapy for resection of liver metastases may have curative potential. Radiofrequency ablation (RFA) alone or in combination with resection as well as stereotactic body radiation may be applied as an additive option in patients with primarily unresectable liver metastasis. Cytoreductive surgery plus intraabdominal chemotherapy (HIPEC) may be applied in patients with limited resectability of peritoneal metastases [3]. Currently, only few clinical trials investigate novel immunotherapeutic approaches in mCRC, while most trials focus on combinations of established drugs and chemotherapy regimens. Personalized peptide vaccination targeting individual tumor-specific mutations has become a promising treatment option for many malignancies. Immunogenicity, feasibility, and toxicity were favorable in several clinical trials [4,5]. Neoantigens may act as potent tumor rejection antigens, given prerequisites such as antigen expression, tumor infiltration by neoantigen-specific T cells, and immune-favorable tumor microenvironment (TME) are provided. Neoantigen-targeting T cells are thought to be the main mediators of CPI efficacy [6].

Here, we report a patient’s case with advanced mCRC who received a personalized neoantigen-targeting peptide vaccine and subsequently showed tumor infiltration of vaccine-induced T cells and OS of 41 months after the initial diagnosis.

## 2. Materials and Methods

### 2.1. Vaccine Design and Vaccination

Neoepitope prediction and vaccine peptide selection were performed as previously described [7,8]. Briefly, tumor variants were phased, and the coding nucleotide sequences were translated into amino acid sequences. The patients’ HLA type was extracted from exome sequencing data and HLA binding affinity was predicted for all resulting peptide sequences harboring amino acid variants. Potential HLA class I neoepitopes (peptide length between 8 and 12 amino acids) to be included in the vaccine were selected according to a scoring algorithm, including predicted binding affinity (using NetMHCpan, NetMHC (DTU Health Tech, Lyngby, Denmark), and SYFPEITHI (Department of Immunology, Tübingen, Germany)), tumor abundance (high variant allele frequency), and high variant expression (tumor transcriptome sequencing). Peptides with binding potential to multiple HLA class I molecules of the patient were preferably selected. Coverage of all patient HLA class I alleles by predicted binding peptides was aspired. Potential HLA class II neoepitopes (peptide length about 17 amino acids) were selected according to tumor abundance (high variant allele frequency) and high variant expression (tumor transcriptome sequencing).

Vaccine peptides were synthesized by solid-phase peptide synthesis (SPPS). Single peptides were purified to reach at least 95% purity (Intavis Peptide Services GmbH, Tübingen, Germany). The lyophilized peptides (HCl salt) were dissolved in water (Aqua ad iniectabilia; BBraun, Melsungen, Germany) containing 33% dimethylsulfoxide (Miltenyi, Bergisch Gladbach, Germany). Peptide solutions of a maximum of 10 individual peptides were combined to a cocktail and sterile-filtered through PTFE-membrane filters (Millex-LG sterile filter; Merck Millipore, Darmstadt, Germany). Peptide cocktails were then bottled in glass vials (Thermo Scientific, Langenselbold, Germany). The final vaccine cocktail batches were controlled for identity and purity of the contained peptides. Sterility and endotoxin controls were performed before final QC/QA release. Vaccine vials were stored at −80 °C until use.

Per vaccination, 0.5 mL multi-peptide solution (0.8 mg/mL per peptide) of each cocktail A and B were injected intracutaneously in the left or right lower abdomen. Vaccine adjuvants were administered by subcutaneous injection of 83 µg Sargramostim and superficial application of Imiquimod (50 mg creme) at the peptide cocktail injection sites.

### 2.2. Immunomonitoring

PBMCs were stimulated with vaccine peptides (1 µg/m for MHC class I peptides and 5 µg/mL for MHC class II peptides) either separately or in pools. Cells were cultivated in the presence of IL-2 (10 U/mL; Miltenyi Biotec, Bergisch-Gladbach, Germany) and IL-7 (10 ng/mL; Miltenyi Biotec) for 12 days. Cultures were split and restimulated for 14 h with peptide or DMSO in the presence of Golgi inhibitors (1 µL/mL; Golgi Plug, BD Biosciences, Franklin Lakes, NJ, USA). A separate positive control contained cells stimulated with an antibody-based non-specific stimulus (10 µL/mL; human CytoStim^TM^, Miltenyi Biotec). All samples were stained extracellularly for CD3, CD4 and CD8 and with Zombie L/D marker for viability (all Biolegend, Amsterdam, The Netherlands). CD4+ and CD8+ T cells were gated within single, viable, CD3+ positive cells. Intracellular cytokine staining for markers IFN-γ, TNF-α, IL-2, and CD154 (all Biolegend) and Boolean gating of marker-positive cells was performed to identify polyfunctional T cells. Frequency and magnitude of peptide-specific T cell responses was calculated by integration of negative control and peptide-stimulated sample Boolean gating values (stimulation index and percentage of specific T cells). Peptides were partly pooled for immunomonitoring due to limited PBMC availability. Samples were measured on a NovoCyte 3005R cytometer (Agilent, Santa Clara, CA, USA). Data were analyzed using FlowJo 10.5.3 (FlowJo LLC, Ashland, AZ, USA).

### 2.3. TIL Isolation

Tumor samples were collected directly after surgery, washed with PBS, and bloody and necrotic areas were discarded. Samples were further cut into small fragments (~1 mm^3^) and transferred to TIL medium (RPMI + Glutamax with 10% human AB serum and 1% PenStrep + 1000 IU/mL IL-2 + 30 ng/mL OKT3/anti CD3 antibody). A total 50% of medium was exchanged every 2–3 days adding fresh IL-2 (but not OKT3). Expansion was conducted for 28 days and TILs were harvested and cryopreserved in serum-free cell freezing medium until further use. For detection of vaccine-specific T cells, the same protocol as for PBMCs was used (see Section 2.2).

### 2.4. Fluorescence-Activated Cell Sorting

PBMCs were stimulated with the same peptide pool that was found positive in TILs. Cells were harvested after 12 days, restimulated with the peptide pool for 14 h with additional anti-CD40 pure functional grade (Miltenyi Biotec). Cells were stained extracellularly for viability (Zombie L/D), CD3, CD4, CD8, and CD154 for cell sorting using the MACSQuant Tyto Cell Sorter (Miltenyi Biotec) system. Cells were sorted on viable/CD3+/CD4+/CD154+ markers. All samples were directly cryopreserved as cell pellets for DNA isolation.

### 2.5. DNA Isolation

DNA was isolated using the QIAamp DNA micro kit (Qiagen, Hilden, Germany) according to the manufacturer’s instructions.

### 2.6. TCRβ CDR3 Sequencing

TCRβ CDR3 sequencing was performed as described previously [8]. Briefly, sequencing libraries were prepared from 100 to 250 ng DNA derived from respective samples, using the AmpliSeq TCR beta kit (Illumina, Berlin, Germany) according to the manufacturer’s instructions. Quality control of libraries included Qubit (Thermo Fisher Scientific, Waltham, MA, USA) and Fragment Analyzer (Agilent) measurement. Libraries were sequenced on an Illumina NovaSeq 6000 sequencer with read length set to 2 × 100 bp. Sequencing reads were demultiplexed using Illumina bcl2fastq (2.20). Adapters were trimmed using Skewer (version 0.2.2). Quality trimming of reads was not performed. Prior to TCR analysis, FASTQ files were downsampled to 2 million read pairs. Paired reads with overlap were merged into single reads. Read pairs that could be successfully merged into single reads were further analyzed. TCR CDR3β regions and TCR sequences were reconstructed using NGmerge and RTCR. All resulting non-functional clones (alternative reading frames, premature stop codons) were discarded from analysis. Finally, TCR clones were annotated with corresponding read counts and frequencies. Clones with less than 100 reads in individual samples were discarded for comparative analysis.

## 3. Results

### 3.1. Case Presentation

A patient in his 50s was first diagnosed with stage 4 mCRC in January 2020. He underwent colectomy and lymphadenectomy in February 2020. The disease was classified as pT4a, N2b (11/29), L1, V1, Pn1, R1, MSS, *KRAS* c.35G>T; p.G12V, *BRAF* WT, *NRAS* WT. Post surgery, he received four cycles of FOLFIRINOX chemotherapy. In June 2020, liver metastases were removed by liver wedge resection of segments 4a and 5 and total resection of segment 7. Afterwards, chemotherapy regimen was switched to capecitabine and oxaliplatin, which was stopped after one cycle due to grade 3 side effects. Chemotherapy regimen was then switched back to FOLFIRINOX (another four cycles). In January 2021, chemotherapy was switched to capecitabine maintenance with additional metformin. Bevacizumab was additionally applied continuously. In April 2021, MRI scans revealed the presence of at least three new liver metastases which remained stable in size until April 2022, when a clear size increase was detected. At the time of detection of the new metastases, a sample of the resected primary liver metastases was used for isolation of DNA and RNA, and exome and transcriptome sequencing were performed with the aim of obtaining a molecular profile and potential therapeutic targets. The calculated TMB was low with 1.9, indicating a pMMR type tumor. The analysis revealed activating missense mutations in tumor drivers *KRAS* (c.35G>T; p.G12V) and *TP53* (c.524G>A; p.R175H), and inactivating mutations in *APC* (c.2413C>T; p.R805*), *ARID1A* (c.6628C>T; p.Q2210*), and *RBM10* (splice site, c.503-2A>T; p.?). The sequencing data was further used for designing a personalized peptide vaccine targeting tumor-specific mutations and vaccination was started within the scope of an individual healing attempt in February 2022. Peptide vaccines were administered according to a prime–boost scheme for a total of 12 vaccinations from February 2022 until April 2023. Each vaccination consisted of intracutaneous injection of two multi-peptide solutions, each containing 10 individual peptides, into the lower abdomen, followed by subcutaneous injection of 84 µg Sargramostim (GM-CSF) and topical application of Imiquimod at the vaccination site. In response to the size increase in liver metastases, FOLFIRI chemotherapy was started in April 2022 and maintained until July 2022, when liver metastases were resected by hemihepatectomy. Tissue samples from two of the resected metastases were used for TIL isolation, exome and transcriptome analysis and TCRβ profiling. In November 2022, further new metastases in the lung and lymph nodes (mediastinal and hilar) were detected and FOLFOX chemotherapy with additional Bevacizumab was started in December 2022 (six cycles). In April 2023, off-label therapy with Lenvatinib and Pembrolizumab was started with the intention to modulate the tumor microenvironment and support vaccine-induced T cells. In June 2023, the disease further progressed with multiple new bone metastases. All therapeutic interventions were stopped by then and the patient unfortunately deceased in July 2023 (OS: 41 months) (Figure 1A).

### 3.2. Vaccine Immunogenicity and Safety

Immunomonitoring revealed multiple vaccine-induced neoantigen-specific CD4+ and CD8+ T cell responses (against at least 10/18 targeted variants) after six vaccinations (three months after vaccination start). CD4+ T cell responses were directed against at least six distinct neoantigens (including the driver mutation *KRAS* (c.35G>T; p.G12V)) and CD8+ T cell responses were directed against at least five distinct neoantigens (Figure 1B). T cell responses were polyfunctional, robust, and persistent. Side effects were moderate and limited to temporal grade 2 injection site reactions.

### 3.3. Tumor Infiltration by Vaccine-Induced T Cells

Tissue samples of the two liver metastases (M1 and M2) resected five months after vaccination start were used for TIL isolation. Immunomonitoring confirmed that neoantigen-specific CD4+ T cells directed against vaccinated peptides were present in TILs derived from M1, but not M2 (Figure 2A).

Next, we identified TCRβ CDR3 sequences of vaccine-induced T cells. For this purpose, we used PBMC samples and stimulated for 12 days with the same peptide pool that was found positive in TILs (same protocol as for immunomonitoring). Peptide-specific CD4+ T cells were bulk sorted according to CD154 expression in response to peptide stimulus. We performed bulk sequencing of TCRβ CDR3 regions in this sample and additionally in both metastases M1 and M2. Comparison of TCRβ clonotypes between all three samples revealed a higher clonal diversity in M1 compared to M2. Strikingly, the clonal overlap of the sorted T cell sample with M1 was higher than with M2, supporting the finding of vaccine-specific T cell infiltration in M1 but not M2 (Figure 2B).

Finally, we performed exome and transcriptome sequencing on both post-vac tumor samples (M1 and M2) and analyzed allele frequencies and expression of vaccine-targeted variants in comparison to the pre-vac tumor sample. Several tumor-specific variants against which the vaccine successfully generated T cell responses were diminished or even absent in the M1 sample, while most variants remained stable or even increased in the M2 sample. Of the five vaccine-targeted variants for which we could show specific T cell infiltration in M1, three exhibited a reduced allele frequency (*GPN2* c.758A>G; p.Q253R, *KRAS* c.35G>T; p.G12V and *MTBP* c.164A>T; p.N45I) and two were not detectable at all (*GGPS1* c.644T>C; p.F215S and *URB2* c.1487C>T; p.T496M) (Figure 2C).

## 4. Discussion

In this report, we describe the clinical course of a patient with mCRC who received a fully personalized neoantigen-targeting peptide vaccine in addition to standard of care. The patient survived for 41 months from his initial diagnosis despite a high initial tumor burden and continuous progressive disease. The vaccine was administered in a prime-boost scheme for a total of 12 vaccinations over the course of 14 months. Side effects were tolerable, and the vaccine exhibited extraordinary immunogenicity, despite concurrent chemotherapy. Furthermore, functional vaccine-induced T cells were shown to be able to infiltrate tumor and possibly eradicate tumor cell clones carrying certain vaccine-targeted variants. Strikingly, 4/18 targeted variants were no longer detectable in T cell-infiltrated tumor tissue obtained after initiation of vaccine therapy. Additionally, several of the targeted variants were diminished in the metastasis that exhibited vaccine-specific T cell infiltration, possibly reflecting immune editing [9]. The selective pressure applied by the presence of antigen-specific T cells may reshape clonal composition and finally lead either to tumor escape or eradication. We can only speculate on the molecular features that allowed the infiltration of vaccine-induced T cells into only one of the metastases. We looked at the presence of regulatory T cells and M1 and M2 macrophages in both tissue samples but found no differences. Furthermore, we looked at expressions of HLA class I and II, CD31, and PD-L1 but found no differences between M1 and M2. T cell infiltration may have been triggered by local inflammation and endothelial activation in only one of the metastases, but we did not analyze differential expression of markers supporting endothelial transmigration like TNF-α, VEGF, CD99, or CD144.

The vaccination approach pursued in this case, compared to other neoantigen vaccines, comprises a rather large number of peptides (20) targeting a high number of neoantigens (18). In addition, the vaccine comprises both potential HLA class I restricted minimal epitopes and longer peptides aiming at potential presentation by both HLA class I and II. While we did not evaluate cross-reactivity of neoantigen-specific T cells to the corresponding wildtype peptides, we did not observe any tissue toxicities in the patient. We successfully applied this vaccination approach in other malignant entities in larger patient cohorts and found the same favorable safety profile [7,8,10]. This is in line with published safety data from clinical trials investigating neoantigen peptide vaccination [4,5,11,12]. Prediction of neoantigen immunogenicity is still largely impossible and neoepitopes are believed to rather be unique than shared among patient groups. This was shown for CRC but is probably true for most cancers and their respective neoantigens [13]. Hence, we propose targeting a broad number of tumor mutations with an individualized vaccine is key to induction of adequate T cell responses and potential clinical efficacy.

Despite induction of durable T cell responses and tumor infiltration, the vaccination alone did not prevent the development of further metastases. This may in part be due to a lack of presentation of vaccine targets by HLA molecules on tumor cells. We excluded loss of general HLA expression in tumor cells by immunohistochemistry. Thus, the lack of neoantigen presentation is more likely linked to cancer cell-intrinsic mechanisms such as reduced or failed production of altered proteins, failed antigen processing, or HLA binding incompatibility [5,14]. When we analyzed composition of T cell infiltrates in metastasis samples by immunofluorescence, we found the presence of CD8+ T cells (about 60% of CD3+ cells exhibited CD8 expression). In contrast, we did not observe vaccine-specific CD8+ T cells in TIL samples. However, we only tested TIL specificity for a subset of vaccine peptides (Figure 2C). Combination with CPI or further TME-modulating agents would be possible, given the favorable safety profile of the peptide vaccine, and may improve clinical effectiveness. CPI alone has not shown convincing results in clinical trials in MSS mCRC, probably due to low TMB and low expression of PD-L1; however, there may be a rationale for combination therapy with peptide vaccination [15]. We analyzed PD-L1 expression in metastases by immunohistochemistry and found low levels of expression with 1–5% of tumor cells staining positive for PD-L1. Combination therapy of peptide vaccination and checkpoint inhibition has shown promising results in clinical trials [11,12]. Only recently, a clinical trial investigated safety and clinical efficacy of a neoantigen-targeting approach with promising first results in mCRC [16]. Targeting neoantigens in combination with checkpoint blockade may constitute a new treatment approach for MSS CRC with the potential to reprogram the TME towards an inflamed phenotype and impacting clinical outcomes [17].

In the present report, we demonstrate the feasibility and immunogenicity of a personalized neoantigen-targeting peptide vaccine in a patient with mCRC. Vaccination induced specific T cell responses against a high number of neoantigens without severe side effects. Nevertheless, a regression of metastases was not observed. Additional comparative analyses investigating factors like actual neoantigen presentation and TME composition in primary tumor and metastasis tissues would have been helpful to explain differential tumor infiltration by TILs and clinical outcome in general. Such extensive analyses were not feasible due to limited sample amounts in this single case. The results discussed in this manuscript originate from treatment of a single patient and general conclusions for clinical benefit cannot be made. Hence, these results warrant for conduction of clinical trials in larger patient cohorts to further investigate the safety and clinical efficacy of this promising treatment approach.

## 5. Conclusions

This case demonstrates immunogenicity of a neoantigen-derived peptide vaccine and highlights tumor-infiltrating capabilities and potential cytotoxicity of vaccine-induced T cells in mCRC.

## Figures and Tables

**Figure 1 vaccines-13-00960-f001:**
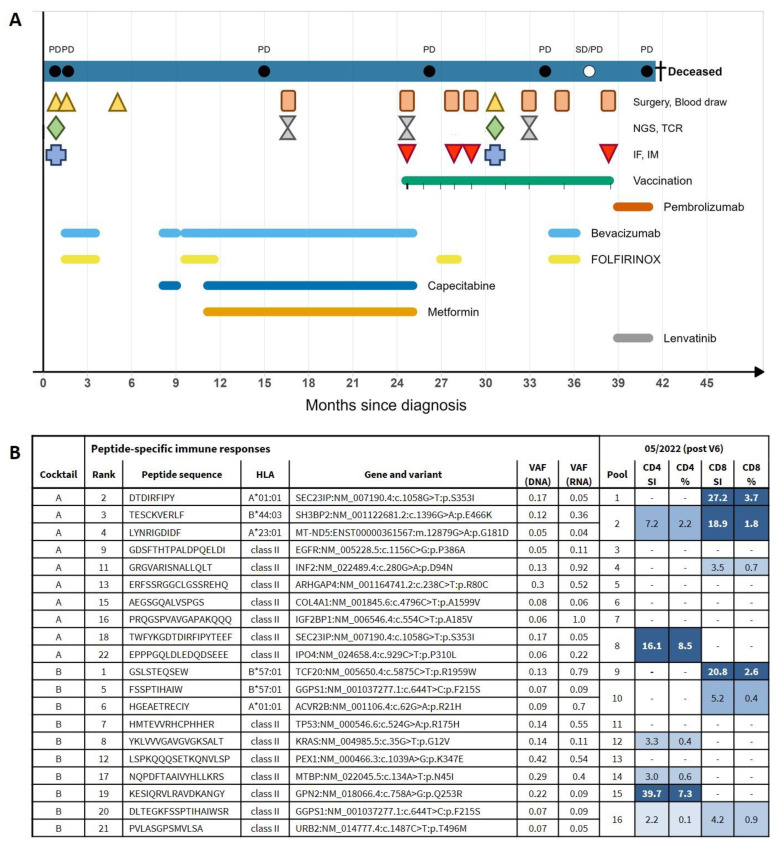
Graphical illustration of clinical course and vaccine immunogenicity. (**A**) Swimmer plot depicting all treatment interventions since initial diagnosis. From top to bottom: Staging (circles), Surgery (orange triangles), Blood drawings (brown squares), Exome and transcriptome sequencing (NGS; green diamonds), TCRβ sequencing (TCR; gray hourglasses), Immunofluorescence (IF; blue crosses), Immunomonitoring (IM; red triangles), Vaccination, other concomitant therapies. SD = stable disease, PD= progressive disease. (**B**) Vaccine composition and immunomonitoring results in a blood sample taken after six vaccinations (three months after vaccination start). Peptides were separated in two vaccine cocktails A and B. Rank column marks peptide rank according to vaccine design pipeline. HLA column gives putative HLA restriction of peptides. Allele frequencies for DNA and RNA of individual variants are annotated. Pool describes peptide pooling strategy for immunomonitoring. Columns “CD4” and “CD8” contain immunomonitoring results as calculated stimulation index (SI) values and percentage of peptide-specific T cells. Color code indicates strength of T cell response. VAF = variant allele frequency, HLA = human leukocyte antigen, SI = stimulation index.

**Figure 2 vaccines-13-00960-f002:**
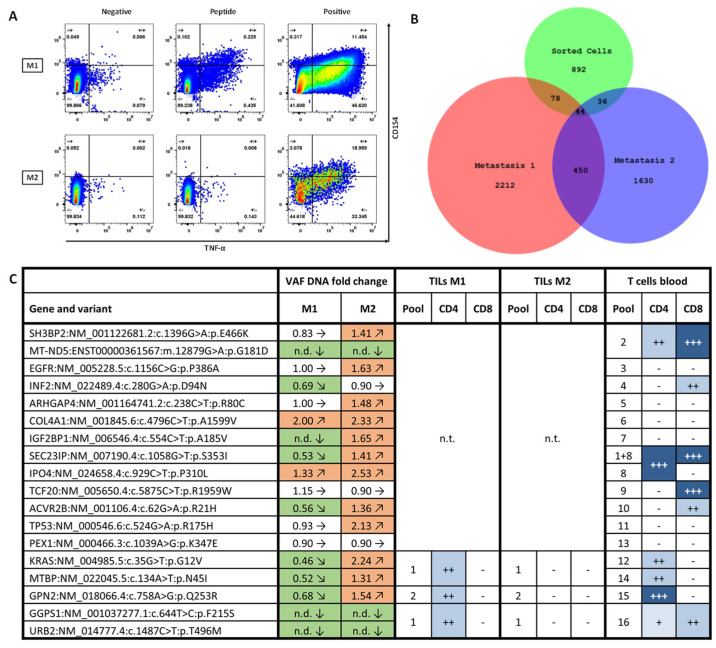
Analysis of tumor metastases under vaccine therapy. (**A**) Immunomonitoring of TILs derived from two distinct liver metastases M1 and M2. TILs were gated for singlets/viable cells/CD3+/CD4+. FACS plots depict CD4+T cell expression of TNF-α (x-axis) and CD154 (y-axis). TILs derived from M1 (upper row) and M2 (lower row) in response to stimulation with DMSO (Negative, left), Peptide pool (Peptide, middle) and Cytostim^TM^ (Positive, right). (**B**) Overlap of TCRβ CDR3 sequences identified in samples of M1 (red), M2 (blue) and a sample of activation marker-based sorted T cells with vaccine peptide pool specificity (sorted cells; green). (**C**) Vaccine-targeted variants and respective fold change in variant allele frequency in M1 and M2 compared to primary tumor sample. Allele frequencies were determined by whole exome sequencing and adjusted for tumor cell content of the respective samples. Color code and arrows resemble increase (red, arrow upwards), decrease (green, arrow downwards) or unchanged (white, arrow sidewards). CD4+ and CD8+ T cell reactivities against variants in blood and TILs as determined by immunomonitoring. Blue color code and amount of “+” resembles the strength of T cell response. White color and “-“ indicates no T cell response was measured. VAF = variant allele frequency, n.d. = not detectable, n.t. = not tested.

## Data Availability

The raw data supporting the conclusions of this article will be made available by the authors on request.

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
