# Peer review of "Case Report: A Multi-Peptide Vaccine Targeting Individual Somatic Mutations Induces Tumor Infiltration of Neoantigen-Specific T Cells in a Patient with Metastatic Colorectal Cancer"

_vaccines, 2025, doi:10.3390/vaccines13090960_

Round 1

Reviewer 1 Report

Comments and Suggestions for Authors

Rabsteyn et al reported a patient’s case with advanced metastatic colorectal cancer (mCRC) who was treated with a neoantigen derived multi-peptide vaccine in addition to standard of care.

It is well-known that predicted peptides may not be processed by cancer cells harboring the mutated antigens. Although, the vaccination induced specific T cell responses against a high number of neoantigens without severe side effects in this manuscript, vaccination did not prevent development of further metastases. Probably due to lacking expression of predicted peptides presented by HLA molecules on cancer cells, therefore, vaccination-induced specific T cells may not be able to recognize and kill cancer cells harboring mutated antigens. Please address such concerns in the section of Discussion.  

Author Response

Comment 1: It is well-known that predicted peptides may not be processed by cancer cells harboring the mutated antigens. Although, the vaccination induced specific T cell responses against a high number of neoantigens without severe side effects in this manuscript, vaccination did not prevent development of further metastases. Probably due to lacking expression of predicted peptides presented by HLA molecules on cancer cells, therefore, vaccination-induced specific T cells may not be able to recognize and kill cancer cells harboring mutated antigens. Please address such concerns in the section of Discussion. 

Answer 1: We thank reviewer 1 for this valuable suggestion to improve the manuscript. The discussion section was amended accordingly: “This may in part be due to a lack of presentation of vaccine targets by HLA molecules on tumor cells. We excluded loss of general HLA expression in tumor cells by immunohistochemistry (data not shown). Thus, lack of neoantigen presentation is more likely linked to cancer cell intrinsic mechanisms such as reduced or failed production of altered proteins, failed antigen processing or HLA binding incompatibility [5,14]. “ reference 14 was added.

Reviewer 2 Report

Comments and Suggestions for Authors

This manuscript presents a case report of a patient with advanced metastatic colorectal cancer treated with a personalized multi-peptide neoantigen vaccine along standard therapies. The study is interesting as it showcases a novel personalized immunotherapy strategy in a metastatic colorectal caner, a context that does not typically respond to immunotherapy. The manuscript is well-organized and well-written, and provides thorough immunological characterization. It convincingly demonstrates that the vaccine elicited neoantigen-specific T cells that traffic to the tumor and lead to a loss or reduction of targeted mutant alleles. The extended survival of the patient (41 months) support the potential benefit and feasibility of this approach. 

However the following issues need to be addressed:

  1. This is a case report of a single patient and general conclusions of clinical benefit cannot be made. Disease progression occurred despite vaccination.
  2. Only one metastasis showed vaccine-specific T cell infiltrations. Why did vaccine-specific T cells infiltrate metastasis M1 but not M2? Additional analysis would be very valuable.
  3. It would be valuable to describe the CD8+ T cell activity in the tumors.
  4. It would be helpful to add if the tumor microenvironment in metastasis expressed  PD-L1 or such markers.
  5. Did the authors evaluate whether T cell responses induced by the vaccine are specific to the mutated peptides and not the wild-type sequence? Please discuss this issue.
  6. The disappearance of certain mutant alleles in the infiltrated metastasis is remarkable. Does it indicate eradication of these tumor cell clones by T cells or outgrowth of other clones? Please discuss.

Author Response

We thank reviewer 2 for the valuable suggestions to improve the manuscript.

Comment 1: This is a case report of a single patient and general conclusions of clinical benefit cannot be made. Disease progression occurred despite vaccination.

Answer 1: A sentence was added to the discussion: “The results discussed in this manuscript originate from treatment of a single patient and general conclusions for clinical benefit cannot be made.”

Comment 2: Only one metastasis showed vaccine-specific T cell infiltrations. Why did vaccine-specific T cells infiltrate metastasis M1 but not M2? Additional analysis would be very valuable.

Answer 2: Sentences were added to the discussion: “We can only speculate on the molecular features that allowed infiltration of vac-cine-induced T cells into only one of the metastases. We looked at presence of regulatory T cells and M1 and M2 macrophages in both tissue samples but found no differences (data not shown). Furthermore, we looked at expression of HLA class I and II, CD31 and PD-L1 but found no differences between M1 and M2 (data not shown). T cell infiltration may have been triggered by local inflammation and endothelial activation in only one of the metastases but we did not analyze differential expression of markers supporting endothelial transmigration like TNF-α, VEGF, CD99 or CD144.”

Comment 3: It would be valuable to describe the CD8+ T cell activity in the tumors.

Answer 3: Immunohistochemistry revealed presence of CD8+ T cells in tumors (median 60% of all CD3+ cells). However, we did not observe any vaccine-specific CD8+ T cells within the TILs (see figure 2C). We added sentences to the discussion: “When we analyzed composition of T cell infiltrates in metastasis samples by immuno-histochemistry, we found presence of CD8+ T cells (about 60% of CD3+ cells exhibited CD8 expression, data not shown). In contrast, we did not observe vaccine-specific CD8+ T cells in TIL samples. However, we only tested TIL specificity for a subset of vaccine peptides (Figure 2C).”

Comment 4: It would be helpful to add if the tumor microenvironment in metastasis expressed  PD-L1 or such markers.

Answer 4: See response to point 2. Additionally, we added one further sentence regarding PD-L1 expression: “We analyzed PD-L1 expression in metastases by immunohistochemistry and found low levels of expression with 1–5% of tumor cells staining positive for PD-L1.”

Comment 5: Did the authors evaluate whether T cell responses induced by the vaccine are specific to the mutated peptides and not the wild-type sequence? Please discuss this issue.

Answer 5: We did not evaluate cross-reactivity to wild-type sequences. From our experience with neoantigen-specific T cells, it is not completely exclusionary that cross-reactivity occurs in some cases. However, until now we never observed tissue toxicities against healthy tissues in vaccinated patients. This is in line with published safety data from clinical trials examining neoantigen peptide vaccines. We added sentences to the discussion: “While we did not evaluate cross-reactivity of neoantigen-specific T cells to the corresponding wildtype peptides, we did not observe any tissue toxicities in the patient. We successfully applied this vaccination approach in other malignant entities in larger patient cohorts and found the same favorable safety profile [7,8,10]. This is in line with published safety data from clinical trials investigating neoantigen peptide vaccination. [4,5,11,12].”

Comment 6: The disappearance of certain mutant alleles in the infiltrated metastasis is remarkable. Does it indicate eradication of these tumor cell clones by T cells or outgrowth of other clones? Please discuss

Answer 6: We added sentences to the discussion: “Additionally, several of the targeted variants were diminished in the metastasis that exhibited vaccine-specific T cell infiltration, possibly reflecting immune editing [9]. The selective pressure applied by presence of antigen-specific T cells may reshape clonal composition and finally lead either to tumor escape or eradication”. A reference on immune editing [9] was added.

Reviewer 3 Report

Comments and Suggestions for Authors

This manuscript describes a case report of great interest. Indeed, the authors have shown the possibility of inducing an immune response in a patient with liver metastasis of colorectal carcinoma.

The main interesting point to this reviewer is the response of the patient and the possibility of identifying the presence of antigen-specific T cells in one of the two metastases found. 

The manuscript shows that an immune response is possible in a late-stage patient and that the heterogeneity of metastases is present.

Some discussion on the features of TIL to localize at the tumor tissue is important to note that the immune response can be effective only if immune cells and the tumor microenvironment (TME) allow the targeting of a tumor at the metastasis site.

The expression of markers involved in endothelial transmigration on lymphocytes, as well as the presence of counter receptors in the specific metastasis, could justify the different response. 

Author Response

Comment 1: Some discussion on the features of TIL to localize at the tumor tissue is important to note that the immune response can be effective only if immune cells and the tumor microenvironment (TME) allow the targeting of a tumor at the metastasis site. The expression of markers involved in endothelial transmigration on lymphocytes, as well as the presence of counter receptors in the specific metastasis, could justify the different response.

Answer 1: We thank reviewer 3 for the valuable suggestions to improve the manuscript. The suggestions overlay with suggestions of reviewer 2 and we added several sentences to the discussion: “We can only speculate on the molecular features that allowed infiltration of vac-cine-induced T cells into only one of the metastases. We looked at presence of regulatory T cells and M1 and M2 macrophages in both tissue samples but found no differences (data not shown). Furthermore, we looked at expression of HLA class I and II, CD31 and PD-L1 but found no differences between M1 and M2 (data not shown). T cell infiltration may have been triggered by local inflammation and endothelial activation in only one of the metastases, but we did not analyze differential expression of markers supporting endothelial transmigration like TNF-α, VEGF, CD99 or CD144.”

Round 2

Reviewer 1 Report

Comments and Suggestions for Authors

My concerns have been addressed, I recommend it for publication as it stands.

Author Response

We are happy that we could adress all concerns of reviewer 1 and thankful for the improvement of the manuscript.